# *Lycium Barbarum* Polysaccharide-Iron (III) Chelate as Peroxidase Mimics for Total Antioxidant Capacity Assay of Fruit and Vegetable Food

**DOI:** 10.3390/foods10112800

**Published:** 2021-11-14

**Authors:** Shuo Shi, Jianxing Feng, Yanmin Liang, Hao Sun, Xuewei Yang, Zehui Su, Linpin Luo, Jianlong Wang, Wentao Zhang

**Affiliations:** College of Food Science and Engineering, Northwest A&F University, 22 Xinong Road, Xianyang 712100, China; shishuo1025@163.com (S.S.); fengjianxing98@nwafu.edu.cn (J.F.); liangyymm@163.com (Y.L.); 15591898565@163.com (H.S.); yangxuewei@nwsuaf.edu.cn (X.Y.); suzehui0809@163.com (Z.S.); linpinluo@nwafu.edu.cn (L.L.); wanglong79@nwsuaf.edu.cn (J.W.)

**Keywords:** total antioxidant capacity (TAC), peroxidase, polysaccharide-iron complex, steady-state kinetics, *Lycium barbarum* polysaccharide (LBP)

## Abstract

Quantitative evaluation of the antioxidant capacity of foods is of great significance for estimating food’s nutritional value and preventing oxidative changes in food. Herein, we demonstrated an easy and selective colorimetric method for the total antioxidant capacity (TAC) assay based on 3,3’,5,5’-tetramethyl-benzidine (TMB), hydrogen peroxide (H_2_O_2_) and synthetic *Lycium barbarum* polysaccharide-iron (III) chelate (LBPIC) with high peroxidase (POD)-like activity. The results of steady-state kinetics study showed that the K_m_ values of LBPIC toward H_2_O_2_ and TMB were 5.54 mM and 0.16 mM, respectively. The detection parameters were optimized, and the linear interval and limit of detection (LOD) were determined to be 2–100 μM and 1.51 μM, respectively. Additionally, a subsequent study of the determination of TAC in six commercial fruit and vegetable beverages using the established method was successfully carried out. The results implied an expanded application of polysaccharide-iron (III) chelates with enzymatic activity in food antioxidant analysis and other biosensing fields.

## 1. Introduction

Reactive oxygen species (ROS) are triggers of oxidative stress (OS) in the human body, which may result in various degrees of oxidative injury to human tissues and organs [1,2]. Thus, oxidative stress mediated by excessive ROS is a noteworthy inducer of chronic diseases such as presenility, cancer, cardiovascular, and neurodegenerative diseases [3]. Fortunately, bioactive antioxidants in foods are able to not only be used as precautions against chronic illnesses but also prevent food from spoiling during storage [4]. Recently, total antioxidant capacity (TAC), a representative index of the antioxidative function of food, has been correlated with food trophic value and storage stability [5]. Analytical methods for the measurement of antioxidant capacity of food samples using large instruments, such as electron paramagnetic resonance (EPR) spectroscopy [6], attenuated total reflectance infrared spectroscopy (ATR-IR) and Raman spectroscopy, have definitely been shown to be effective [7]. However, the shortcomings of requiring expensive instruments and complicated operations limit their use. To meet the need for the rapid determination of the antioxidant capacity of samples, colorimetric methods with a rapid detection speed, low cost and simple operation have been commonly used. Specifically, previous reports have demonstrated several reliable spectrophotometric TAC methods, such as the 2,2′-diphenyl-1-picrylhydrazyl (DPPH) radical assay [8], 2,2-azinobis-(3-ethylbenzothiazoline-6-sulfonate) (ABTS) radical assay [9], Folin–Ciocalteu (FC) assay and oxygen radical absorbance capacity (ORAC) assay [10,11]. Nevertheless, the use of serial complex chemical reactions inevitably results in the drawback of poor repeatability. Hence, the establishment of a convenient method with a high accuracy for TAC detection still needs to be explored.

In recent years, artificial mimetic enzymes with oxidase activity or peroxidase (POD)-like activity, particularly nanozymes developed from functionalized metals or metal oxides, have been used to construct multifarious techniques in biosensing [12]. Inspired by the active site composition and possible catalytic mechanism of horseradish peroxidase (HRP) [13], iron-based mimic enzymes, especially iron-based nanozymes, were constructed as artificial peroxidases. According to earlier reports, nanozymes based on iron oxides such as Fe_2_O_3_ or Fe_3_O_4_ were perceived to be dependable, low cost and simple in preparation [14,15]. Nonetheless, the disadvantages of toxicity evoked by free radicals generated during the catalytic process, limitation for the catalytic condition and poor selectivity of iron-based nanozymes fettered their development in the area of catalysis [16]. Thus, to make iron-based nanozymes available for exploitation, strategies such as organic modification for basic iron oxide nanozymes, construction of single-atom iron nanozymes and ferrous metal-organic frameworks (MOFs) have been implemented, which enhance the catalytic properties of nanozymes or overcome certain shortcomings of toxic effects or strict catalytic conditions in applications [17,18,19]. However, the modification methods mentioned above are usually not only costly but also complicated to operate; hence, more efficient solutions still need to be put forward.

To establish a handy, accurate and reliable TAC assay by means of designing and synthesizing a novel artificial enzyme overcoming the major drawbacks of traditional iron-based artificial enzymes, based on the internal sheath-core structure of iron-dextran [20], we propose a distinctive strategy to synthesize a polysaccharide-iron complex as an organic modified iron-based artificial enzyme. Polysaccharides are usually considered (1) stabilizers for other materials, (2) toxicity reduction media, (3) control agents for the growth of some nanoparticles, (4) molds for specific oxide structures and (5) carriers in functional materials [21]. In this study, *Lycium barbarum* polysaccharide (LBP) with favorable natural activities [22] was used to chemically synthesize *Lycium barbarum* polysaccharide-iron (III) chelate (LBPIC) through a self-embedding process. The results of the characterization of the product demonstrated the successful synthesis of LBPIC, and the catalytic test results and study of the steady-state kinetics of LBPIC showed that to establish a new TAC assay with applicable reliability, low cost and reasonable accuracy, the prepared LBPIC was used as a peroxidase mimic, and ascorbic acid (AA) served as an antioxidant model [23]. Finally, the TAC assay was conducted to quantitatively evaluate the TAC in commercial beverages, and the results demonstrated that the proposed TAC assay had a short test time, high feasibility, accurate quantification and strong resistance to interference.

## 2. Materials and Methods

### 2.1. Material and Reagents

Crude *Lycium barbarum* polysaccharide was purchased from Shanghai Yuanye Bio-Technology Co., Ltd. (Shanghai, China). Graphene oxide powder was obtained from Nanjing Xianfeng NanoMaterial Technology Co. Ltd. (Nanjing, China). 3,3’,5,5’-Tetramethylbenzidine (TMB) and NaBr (FT-IR grade) were purchased from Shanghai Aladdin Biochemical Technology (Shanghai, China). H_2_O_2_ and ascorbic acid were obtained from Guangdong Guanghua Technology Co., Ltd. (Guangdong, China). Other reagents were all obtained locally. All experimental aqueous solutions were prepared with distilled water.

### 2.2. Purification and Decoloration of Crude LBP

Generally, crude LBP was purified using a water solution and alcohol sedimentation method. Then, the decoloration of LBP after purification was carried out using prepared aminated reduced graphene oxide (NH_2_-rGO), which was synthesized by the method of Shi et al. [24]. Specific process details are given in Appendix A of the Appendix A.

### 2.3. Synthesis of LBPIC

Briefly, LBPIC was synthesized based on the method used by Zhang et al. [25]. Trisodium citrate (0.4 g) was completely dissolved in the decolored LBP aqueous solution. FeCl_3_ (2 M) was added dropwise with continuous stirring until the solution remained turbid, and the pH value was adjusted to 7.0–8.0 using NaOH (2 M). The solution was heated in a water bath at 70 °C for 1 h. After centrifugation (5000 rpm, 5 min), the precipitate was dissolved in a small amount of water and washed with alcohol at least three times. Finally, the product was lyophilized, and LBPIC was the resulting reddish brown powder.

### 2.4. Characterization of LBPIC

The prepared LBPIC was characterized by scanning electron microscopy, FT-IR spectroscopy, X-ray diffraction and X-ray photoelectron spectroscopy. The physical properties of LBPIC and storage stability of its aqueous solution at room temperature were recorded by means of visual inspection (Appendix A of the Appendix A).

### 2.5. Peroxidase-Like Activity of LBPIC

Peroxidases, such as HRP, are able to catalyze H_2_O_2_ to produce •OH radicals, which can oxidize colorless TMB to blue ox-TMB. In summary, the POD-like activity of LBPIC was assessed through a typical TMB color change experiment. The experimental group contained acetate buffer (0.01 M, pH 3.0), H_2_O_2_ (25 mM), TMB (1 mM) and LBPIC (25 μg/mL), while the control group contained TMB and H_2_O_2_ or only TMB. After incubation for 5 min at room temperature, the absorbance of each group was measured on a UV–vis spectrophotometer (UV-2550, Shimadzu, Japan). The experiments for optimization of the reaction conditions, including pH, temperature, and time-dependent kinetics, were conducted via the methods outlined in Appendix A of the Appendix A.

### 2.6. Enzyme Kinetics Studies of LBPIC

A steady-state kinetics assay was performed in acetate buffer (0.01 M, pH 3.0) by changing the concentrations of H_2_O_2_ (10–240 mM) with a fixed concentration of TMB (1.0 mM) or changing the concentrations of TMB (0.1–1.0 mM) while maintaining the concentration of H_2_O_2_ (25 mM). The absorbances at 652 nm were recorded during reactions. The initial velocity of each catalytic reaction was determined in terms of the gradient of system absorbance variation. Afterward, the data were fit with the Michaelis–Menten equation based on the initial velocities and substrate concentrations to obtain the Michaelis constants K_m_ and V_max_.

### 2.7. Detection of AA

As a representative antioxidant, ascorbic acid can be used to establish indirect detection methods for the total antioxidant capacity of foods [26]. First, the acetate buffer solutions (0.01 M, pH 3.0) that contained H_2_O_2_ (25 mM), TMB (1.0 mM) and LBPIC (25 μg/mL) were mixed with a variety of concentrations of AA (0–100 μM). The absorption spectra of the systems were measured after incubation at room temperature for 180 s using a UV–vis spectrophotometer.

### 2.8. Selectivity Test of the AA Detection Assay

For assessing the selectivity for AA in the detection assay, some substances that commonly occur in food, including Mg^2+^, Ca^2+^, Zn^2+^, K^+^, Na^+^, citric acid (CA), fructose, sucrose, alanine, glutamic acid, aspartic acid, and glycine, were added in sufficient amounts into the AA detection system, with concentrations of these interference factors 20 times that of AA. The absorbances at 652 nm of all test groups were recorded after reaction for 180 s at room temperature.

### 2.9. TAC Detection in Food Samples

The AA detection assay outlined above was validated by measuring the TAC of two vitamin C tablets. Whereafter, this method was used to determine the TAC of five kinds of commercial fruit-vegetable beverages and one fermented beverage. First, the beverage sample was diluted so that the determination results could be determined from the linear range of AA detection. Afterward, the diluted samples were used in place of the AA. Hence, the absorbances of the detection system were converted into the micromolar equivalent of AA. Ultimately, combining the dilution ratio of the sample and the calculation result from the colorimetry, the TAC of the food sample was expressed in the form of AA equivalents. 

## 3. Results and Discussion

### 3.1. Characterization of LBPIC

In our study, as depicted in Figure 1a, crude LBP was amply dissolved in distilled water, filtered and precipitated with aqueous ethanol (ethanol content: 80%) to remove multiple water-insoluble or alcohol-soluble impurities. After the separation and lyophilization of polysaccharide deposits, LBP was redissolved, and its decoloration was realized using a nondestructive method where the pigments were efficiently absorbed by prepared NH_2_-rGO. The decolored filtrate of LBP appeared nearly achromatous and transparent, indicating an adequate removal of pigments and impurities from crude LBP.

The raw material LBP aqueous solution was mixed with trisodium citrate, an effective adjuvant for the chelation reaction, to synthesize LBPIC through a self-embedding method with slight modification. Appendix A shows that the product was a water-soluble reddish brown amorphous powder whose aqueous solutions were yellow or brown. To investigate the basic stability of LBPIC, aspects of its water solution were regularly recorded by visual inspection. As shown in Appendix A, the water solution of LBPIC (1.0 mg/mL) had a favorable optical stability during a one-week preservation at room temperature, which revealed the creation of a stable functional polysaccharide-iron chelation.

The possible synthesis mechanism between iron and LBP is shown in Figure 1b. First, a polymeric iron core is formed via the polymerization of ferric ions in a mildly alkaline solution. The iron core then polymerizes with citrate ions to form a ferric citrate complex and citric acid is released during this period. Polysaccharides (LBPs) act as ligands and finally chelate ferric citrate complexes on their surface [27]. The chelating reaction between LBP and iron is terminated by stopping the addition of FeCl_3_ solution when a precipitate emerges and the system remains turbid after stirring, which indicates that the LBP material has been completely consumed to coordinate with the iron.

SEM was used to obtain detail about the morphology of LBPIC. As shown in Figure 1a, the surface of LBPIC appeared sheet-like, irregular and full of dendritic and rod-like structures, which is consistent with the study by Zhang et al. [28]. Notably, from the SEM results and possible synthesis mechanism of LBPIC, we can conclude that the ferric citrate complex was likely to be wrapped by the LBP. As a consequence, many rod-like microstructures were observed in the morphology of LBPIC, which lends credence to the aforementioned synthesis mechanism.

FT-IR spectroscopy was performed to verify the synthesis of LBPIC. As shown in Figure 1b, the FT-IR spectra of LBP and LBPIC appeared to be similar, and several primary absorption peaks merely shifted to lower wavenumbers. The absorption band of LBP at 3421.7 cm^−1^, which is assignable to the stretching vibration of hydroxyl groups, shifted to 3417.8 cm^−1^ in the spectra of LBPIC [29]. The peaks of LBP at 1626.0 and 1072.4 cm^−1^ related to carboxyl groups shifted to 1618.3 and 1066.6 cm^−1^ in LBPIC, respectively, suggesting the involvement of carboxyl groups in the chelation [30,31]. In addition, the peaks of LBP at 2976.2, 2929.9, 1705.1 and 1051.2 cm^−1^ distinctly vanished in the spectra of LBPIC. The bands at 2976.2 and 2929.9 cm^−1^ corresponded to C—H asymmetric stretching vibrations, while the band at 1705.1 cm^−1^ was related to aldehyde carbonyl groups [32,33]. The band at 1051.2 cm^−1^ was possibly linked with the crystalline phases of LBP [34]. In addition, a broad shoulder occurred at 553.6—474.5 cm^−1^ in the fingerprint region of LBPIC compared to LBP, which might be a result of the significant enhancement of absorption peaks associated with iron oxides [32]. The information above indicated that the added iron ions tended to form iron oxides and that the coordination reaction between iron oxides and polysaccharides mainly occurred on the carboxyl groups, carbonyl groups and hydroxyl groups of LBP, which is in accordance with the idea that the iron in LBPIC exists in the form of iron oxides and that LBP serves as a ligand for the iron core complex.

To determine the concrete states of the surface constituent elements of LBPIC, the XPS spectra of LBPIC were analyzed. First, based on the peaks in Appendix A, it was determined that LBPIC mainly consisted of C, O, N and Fe. A previous report on the monosaccharide formation of LBP validated the existence of mannose, rhamnose, xylose, galactose and glucose in the chemical constitution of LBP [35]. In addition, amino residues are also involved in the glycoconjugates, which contributed to the peaks of C 1s at 285.0 eV, O 1s at 530.9 eV, N 1s at 399.5 eV and Fe 2p at 711.2 eV. As shown in Figure 1c, there were five principal peaks in the high resolution XPS spectrum of Fe 2p. One satellite peak was observed at 719.0 eV, the other was at 733.0 eV, and the binding energy peaks at 711.0 eV and 711.5 eV, which belong to Fe 2p^3/2^, corresponded to FeOOH [36,37]. Then, the peak at 724.7 eV, attributed to Fe 2p^1/2^, was assigned to the characteristic peaks of iron (III) [38]. Moreover, two peaks at 529.9 eV and 531.3 eV were showed in the O 1s XPS spectrum (Figure 1d), which were due to Fe—O—Fe and Fe—O—H, respectively [39]. This result, together with the phenomenon of enhancement of peak absorption in the FT-IR spectra fingerprint region of LBPIC, is corroborative evidence of the existence of iron oxides in LBPIC. According to these results, a chelate composed of LBP and iron oxides was synthesized as expected.

To investigate the compositions and microstructures of LBPIC, XRD was performed, and the results are displayed in Appendix A. There were no sharp peaks in the pattern, indicating that LBPIC was classified as an amorphous material, which is in accord with its appearance. The weak diffraction peaks at 33.5° and 56.0° were ascribed to FeOOH, which confirmed the existence of FeOOH in LBPIC, together with the signals corresponding to FeOOH in XPS. The weak peak at 43.3° was attributed to a compound with iron (III) [40,41]. In certain circumstances, the bonding reaction between iron (III) and the hydroxyl groups of polysaccharides is able to weaken the hydrogen bonds in polysaccharides, leading to the poor absorption peak phenomena of polysaccharide-iron (III) complexes [42]. In addition, the molecular weights and components of polysaccharide material also affect the crystallinity of polysaccharide-iron (III) complexes. As mentioned above, the LBP material is composed of at least five kinds of monosaccharides. LBPIC, with poor homogeneity in its polysaccharide composition, is unlikely to crystallize [25]. All these characterization results demonstrated that an original LBPIC was successfully synthesized via a chemical method.

### 3.2. Peroxidase-Like Activity of LBPIC

To investigate the peroxidase-like activity of LBPIC, a typically used TMB/H_2_O_2_ colorimetric assay was performed at room temperature. TMB is a chromogen extensively used in color-developing methods based on peroxidases. Under acidic conditions, colorless TMB was oxidized to blue ox-TMB in the presence of H_2_O_2_, resulting in an increase in the maximum absorbance at 652 nm (Appendix A). The POD-like activity of LBPIC is presented in Figure 2a. After reaction for the same period of time, H_2_O_2_ was catalyzed to generate •OH radicals by LBPIC, which gave rise to the distinct increase in system absorbance [43]. By comparison, the absorbance change of H_2_O_2_ with TMB or TMB alone was minor. Of note, an earlier report showed that organic modified FeOOH had an excellent peroxidase activity [44], and the existence of FeOOH in LBPIC was also certified by this characterization. Therefore, we could infer that LBPIC possessed prominent POD-like activity, which was endowed by its inner FeOOH in all probability.

Figure 2b shows that the reaction rate was accelerated as the concentration of LBPIC in the system increased. Clearly, the material dosage of 25 μg/mL allowed for both a high reaction velocity and economization of the material. Previous studies have shown that the catalytic activity of enzymes usually relies on pH and temperature. Consequently, multiple variations of the reaction environment, including buffers with different pH values (2–7), different temperatures (20–70 °C) and time-dependent absorbance changes of different dosages of LBPIC (0–45 μg/mL), were assessed to optimize the parameters of the subsequent enzymatic chromogenic reactions with the participation of LBPIC. As shown in Figure 2c,d, the absorbance value of each reaction system was closely associated with the determinate pH value, suggesting a distinct variation trend of the enzymatic activity along with the stated pH value. The enzymatic activity of LBPIC increased monotonically as the pH decreased, while the activity increased stepwise as the temperature rose. Obviously, there was no occurrence of the turning point on the curves, which revealed that the catalytic activity of LBPIC did not exhibit representative inflexion within the established range of conditions.

Notably, TMB seemed to turn green after incubation. To the best of our knowledge, the oxidation of the TMB substrate was conducted under fairly strong acidic conditions in most cases, and the color of TMB was correlated with its structure. In the optimization test for pH, the amino group on one side of TMB was oxidized when the pH value of the back buffer was in the range of 3.0–7.0, which resulted in the generation of blue ox-TMB and presented a linear relationship between the concentrations of ox-TMB and the absorbance values at 652 nm of the corresponding reaction systems. However, if the test was performed in buffer whose pH was below 1.0, the amino groups on both sides of TMB would be affected, and the resultant TMB in diamine form would lead to a yellow color change in the mixture [45]. With the test pH value regulated to 2.0, ox-TMB and the diamine form of TMB were blended after the chromogenic reaction in the system to yield a reaction mixture that was seemingly green, a hybrid color of blue and yellow. It was not clear whether the conversion of ox-TMB to the diamine form of TMB influenced its absorbance at 652 nm, which might cause deviation in the calculation. Consequently, in consideration of the color transition of TMB at pH 2.0, the conditions of the POD-like reaction were set at pH 3.0 and room temperature after optimization, both for the accuracy and convenience of experiments.

### 3.3. Steady-State Kinetics

To obtain the kinetic constants of LBPIC, including K_m_ and V_max_, a steady-state kinetics study was carried out by varying the concentration of substrate TMB under a constant H_2_O_2_ concentration or vice versa at room temperature.
v = V_max_ × [S]/(K_m_ + [S])(1)

In the Michaelis–Menten equation, shown above, v is the initial reaction velocity and [S] is the concentration of substrate. The absorbance data were transformed into concentrations according to the molar absorption coefficient of ox-TMB (39,000 M^−1^cm^−1^).

Typical Michaelis-Menten curves and Lineweaver–Burk double reciprocal plots are shown in Figure 3a–d, and the enzymatic kinetic constants K_m_ and V_max_ were calculated. The K_m_ of an enzyme is closely linked with its affinity to the substrate, and an enzyme with stronger affinity to its substrate usually has a smaller value of K_m_. After calculation and fitting, the K_m_ values of LBPIC toward H_2_O_2_ and TMB were 5.54 mM and 0.16 mM, respectively, and the corresponding V_max_ values were 35.79 × 10^−8^ mM·s^−1^ and 32.21 × 10^−8^ mM·s^−1^, respectively. Compared with HRP, LBPIC had a significantly higher affinity for TMB, and the maximum catalytic reaction velocities of LBPIC with H_2_O_2_ and TMB were apparently higher than that of HRP. Furthermore, the affinity of LBPIC for TMB was also higher than that of some metal-derived and carbon-derived nanozymes (Appendix A). The results of these experiments affirmed that LBPIC was competent as a safe artificial peroxidase.

### 3.4. Colorimetric Detection of AA

AA, which is one of the antioxidants that commonly occurs in foods, is not only used to extend the shelf life but also to guarantee the pleasing presentation, unique flavors and delicate mouthfeel of foods. The blue ox-TMB can be reduced to colorless TMB by AA due to its reducibility. Here, a colorimetric detection assay for AA based on the POD-like activity of LBPIC was successfully constructed. The test conditions were determined to be pH 3.0 and room temperature, and the reaction systems were incubated for 180 s after the addition of a series of concentrations of AA.

Subsequently, as shown in Figure 4a,b, with the increase in the concentration of additional AA, the bleaching degree of colorimetric systems increased accordingly. After linear fitting of the data, there was a remarkable linear negative correlation (R^2^ = 0.99909) between the absorbance of the mixture at 652 nm and the concentration of AA, and the linear range was 2–100 μM. In addition, the LOD (limit of detection) of this method was 1.51 μM, referring to the 3S/N (signal/noise) equation. Compared with some previous AA detection assays based on nanozymes, the TMB/H_2_O_2_/LBPIC system provided a preferable LOD and linear range, and the incubation time of the assay was significantly shorter than that of other methods (Appendix A), which could dramatically accelerate AA determination for food samples. 

The constituent analysis of most food samples is usually regarded as tedious work due to the complicated ingredients of food itself, which have a nonnegligible impact on the detection signal. Therefore, the analytical method for food samples ought to be capable of withstanding certain interferences. To further explore the selectivity of the established colorimetric detection assay for AA, five kinds of metal ions (Mg^2+^, Ca^2+^, Zn^2+^, Na^+^ and K^+^), four kinds of amino acids (alanine, glutamic acid, aspartic acid and glycine), two kinds of disaccharides (fructose and sucrose) and citric acid were used as interference factors. The selected interference factors at given concentrations were added to the chromogenic system, while the control group was carried out by adding AA.

As illustrated in Appendix A, the ox-TMB in the control group was completely bleached. The influences on the color system caused by the interference factors listed were not above 5% according to the comparison between the absorbances of the blank group and the interference test groups. The chromogenic results of test groups were not visibly affected, even though the concentrations of interference factors were 20 times that of AA. In short, the colorimetric assay based on the TMB/H_2_O_2_/LBPIC system has the features of rapid detection, a wide detection range and good selectivity.

### 3.5. TAC Assay

As an indicator to evaluate the nutritional value of food, the TAC, which was determined by the colorimetric AA detection assay, was expressed as micromoles of AA/L. In this study, five kinds of commercial compound fruit and vegetable beverages and one fermented beverage of kiwi, both cloudy and clear types, were selected as testing samples. The compound beverages, including tomato and strawberry, orange and carrot, and grape and blueberry, represented cloudy-type juice, while the other two compound beverages represented clear-type juice. Before the analysis, each beverage was diluted to an appropriate concentration with distilled water to meet two fundamental criteria: (1) the diluted sample was approximately colorless and transparent for the purpose of excluding the interferences resulting from pigments and insoluble beverages, which might make a difference in the results of colorimetric determination; (2) the absorbance of the reaction mixture after chromogenesis was distributed within the linear range of the AA detection method. As shown in Figure 4c, to verify the accuracy of the TAC assay, the TAC of two vitamin C tablets was measured. The test results were in accordance with the AA content in the vitamin C tablets, demonstrating that the TAC assay proposed by us can be used for practical samples on the basis of its high accuracy, good selectivity and convenient operation [46].

As shown in Figure 4d, the results of the test groups revealed that the TAC values of the orange & carrot and grape & blueberry beverages were significantly higher than those of the pineapple & lemon and grape & rose beverages, illustrating that the antioxidant capacity of compound fruit and vegetable beverages of the cloudy type were generally greater [47]. To summarize, the TAC assay based on the TMB/H_2_O_2_/LBPIC system was shown to be feasible, which expands the use of chelates based on polysaccharides in food analysis.

## 4. Conclusions

In summary, crude polysaccharide from *Lycium barbarum* was purified and decolored, and processed *Lycium barbarum* polysaccharide, sodium citrate and ferric trichloride were used to prepare a novel *Lycium barbarum* polysaccharide-iron (III) chelate through a convenient and energy-efficient synthesis procedure. Afterward, the product was characterized, and its peroxidase-like activity was studied by means of the TMB/H_2_O_2_ chromogenic system. LBPIC displayed superior peroxidase-like activity with a K_m_ of 5.54 mM for H_2_O_2_ and 0.16 mM for TMB. Finally, a TAC assay with a wide linear range of 2–100 μM was built on the basis of the TMB/H_2_O_2_/LBPIC system, and AA was used as a practical antioxidant intermediary. In addition, the TAC values of six different commercial beverages were successfully assessed via the constructed colorimetric method. LBPIC, with the advantages of high POD-like activity, high biological security, mild preparation conditions and easily obtained materials, demonstrates good prospects for the fields of food analysis and organic biosensors and also forms the basis of an original application for polysaccharide-iron complexes.

## Data Availability

The data presented in this study are available on request from the corresponding author.

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
