# Peer review of "Lycium Barbarum Polysaccharide-Iron (III) Chelate as Peroxidase Mimics for Total Antioxidant Capacity Assay of Fruit and Vegetable Food"

_foods, 2021, doi:10.3390/foods10112800_

Round 1

Reviewer 1 Report

Dear Authors,

Please find my corrections in the pdf files (manuscript and supplementary file). I believe that the manuscript provided useful information about the use of LBPIC in the field of the food industry. Also, methods and results are presented logically and accurately.

Final decision – minor revision.

Kind regards,

Reviewer

Reviewer 2 Report

The article is very well structured and documented, methods, results and discussion are robust and clearly presented. The language is more than sufficient.

The article is endowed with novelty, originality and high practical interest. 

I suggest only minor changes as follows:

  • Lines 13-14: please explain the abbreviations.
  • Line 209: please cite the "previous report".
  • Line 247: "organic modified FeOOH had with". Change "with" to "an".
  • Lines 269-370: "the antioxidant capacity of compound fruit and vegetable beverages of the cloudy type were generally greater." A reference, if any, confirming this conclusion would be useful.

Reviewer 3 Report

This study proposes the utilization of Lycium barbarum polysaccharide-iron(â…¢) chelate as peroxidase mimics for total antioxidant capacity assay. The research work topic is important and worth of investigation and approving, however there are some shortcomings which must be rectified. In general, important information is presented.  The topic is original. The written can be improved.

  1. The title should be modified respecting the content of the manuscript

     2. This works has a variety of data that are not apparent by just reading the abstract.

  1. In the introduction, author would have benefited from a better understanding of the existing literature.
  2. Please move figure 1 after the paragraph L255-259
  3. L218-L221: please reformulate the sentence
  4. In the discussion, the author would have benefited from a better understanding of the existing literature.
  5. In some sentences, English appears not to be adequate.
  6. To use a space between the number and the unit, as 20 °C; and not to use a space between number and percentage, as 10%, for example.
  7. References must be revised.
  8. Please include a list of abbreviation
